# Syndrome of Transient Headache and Neurologic Deficits with Cerebrospinal Fluid Lymphocytosis (HaNDL): HHV-7 Finding in Cerebrospinal Fluid Challenges Diagnostic Criteria

**DOI:** 10.3390/pathogens12030476

**Published:** 2023-03-17

**Authors:** Anna Sundholm, Rasmus Gustafsson, Virginija Karrenbauer

**Affiliations:** 1Department of Clinical Neuroscience, Karolinska Institutet, 17177 Stockholm, Sweden; rasmus.gustafsson@ki.se (R.G.); virginija.karrenbauer@ki.se (V.K.); 2Department of Neurology, Karolinska University Hospital, 17177 Stockholm, Sweden

**Keywords:** case report, chemokine (C-X-C motif) ligand 13 (CXCL13), corticosteroids, ICHD (3rd ed.), HHV-7, neurofilament light (NfL), syndrome of transient headache and neurologic deficits with cerebrospinal fluid lymphocytosis (HaNDL), valproic acid

## Abstract

The syndrome of transient headache and neurologic deficits with cerebrospinal fluid lymphocytosis (HaNDL) is a rare, self-limiting condition with severe headaches combined with neurological symptoms. However, evidence-based recommendations on diagnostics and treatments are unavailable due to the condition’s rarity and unknown pathophysiology. A young man experiencing severe headache attacks fulfilled the HaNDL diagnostic criteria according to the third edition of the International Classification of Headache Disorders (ICHD-3). We present the dynamics of cerebrospinal fluid (CSF) biomarkers related to low human herpesvirus 7 (HHV-7) load and anti-inflammatory treatment outcomes. Low HHV-7 load may be an immunological trigger of HaNDL, such that elevated levels of CSF- chemokine (C-X-C motif) ligand 13 open a new way to interpret the role of B cells in HaNDL pathogenesis. We discuss the diagnostic challenge of HaNDL, according to the ICHD-3, in the case of pathogen presence at low load in CSF.

## 1. Introduction

The syndrome of transient headache and neurologic deficits with cerebrospinal fluid lymphocytosis (HaNDL) is a rare, self-limiting condition with often severe and alarming symptoms of headache combined with neurological symptoms resulting in the need for emergency care. The pathophysiological aspects behind this disorder are still unknown, and robust best practice regarding diagnostics and treatments is unclear, given the condition’s rarity. However, various explanations regarding pathophysiological aspects have been proposed, including an immune-mediated reaction after viral infection [1]. A review of HaNDL cases has reported a viral-like illness up to three weeks prior to the development of symptoms in 25–40% of cases [2,3]. Moreover, some cases have been described with a possible viral illness and laboratory findings during evaluation for HaNDL syndrome of human cytomegalovirus (hCMV) [4] and human herpesvirus (HHV)-6 in serum [5] and HHV-7 in CSF [1,6]. In this report, we describe the case of a HaNDL-like illness with HHV-7 DNAemia in CSF in an immunocompetent adult.

HHV-7 was first isolated in 1990 from activated CD4+ cells in healthy individuals [7] Like many other members of the HHV family, it is shed in the saliva that constitutes a major transmission route [8]. Even though up to 90% of all adults have detectable HHV-7 DNA levels in their saliva (compared to HHV-6A and -6B, that is present in around 50% of adults) [9], HHV-7 is acquired later in childhood, before 10 years, and at a lower rate than HHV-6A and -6B [10]. After the primary infection, HHV-7 can establish latency in lymphoid cells, just like hCMV and HHV-6A and -6B, from where it can reactivate. Given the high frequency in saliva, it is evident that HHV-7 reactivation occurs very frequently. In contrast, varicella–zoster virus (VZV), another member of the HHV family, typically reactivates once in a lifetime [11]. Like all other HHVs (herpes simplex virus (HSV) 1 and 2, VZV, EBV, hCMV, HHV-6A, and 6B, Kaposi’s sarcoma virus), HHV-7 also exhibit neurotropism [12].

## 2. Case Presentation

The previously healthy 23-year-old male with a history of migraine on his father’s side presented to the emergency department (ED) with aphasia and headache. The day of symptom onset and admission to the ED was defined as day 1. Prehospital clinical signs included anxiety, dysphasia, and numbness in one hand the same day, with a headache of varying intensity for three days. In the ED, the patient was afebrile with normal vital signs. He presented with expressive aphasia during an otherwise normal neurological and general examination. No visible blisters, ulcers, or rashes in the mouth or skin were detected. Blood gas, blood cell count, and C reactive protein (CRP) were all normal, as well as the metabolic panel (blood glucose, kidney, thyroid, and liver function; data are not shown). There was no explanation for his symptoms in the urine toxicology screening and head computed tomography (CT) inclusive angiography (Appendix A). However, lumbar puncture showed increased cerebrospinal fluid (CSF) pressure, increased CSF albumin levels, and vigorously increased numbers of leukocytes (Table 1).

The patient was treated with acyclovir and ampicillin under a working diagnosis of encephalitis and was afebrile the day after with only a slight headache and aphasia as remaining symptoms. Additional extensive blood and CSF testing for possible explanations for cerebrospinal inflammation and/or infection were all negative (Appendix A). Borrelia IgG-antibodies were positive in both CSF and blood, indicating a previously undergone Borrelia infection, so treatment with doxycycline was given as a precaution. During the following days, no increase was seen in either CRP or sedimentation rate. Brain magnetic resonance tomography showed no pathology. Electroencephalogram (EEG) revealed deviations from the normal appearance but no epileptic activity (Appendix A). The patient was mainly afebrile to sub-febrile. During hospitalization, the patient experienced two additional attacks at night with severe headache, aphasia, and confusion. Since the investigation of differential diagnoses remained negative with the symptoms and CSF findings fulfilling the International Classification of Headache Disorders, third edition (ICHD-3) criteria of HaNDL, this diagnosis was given (Appendix A). Treatment with intravenous Solu-Medrol (1 g intravenous on three consecutive days) gave good pain relief, and the patient was discharged from the hospital. The patient experienced a relapse 16 days later after waking up with numbness in his hands and around the mouth and, after an hour, started experiencing a severe headache. In the ED, he had aphasia and suspected left arm weakness, agitation, and restlessness. Headache intensity, anxiety, and agitation escalated with difficulties cooperating with the clinicians. The patient was treated with morphine and diazepam. Due to the escalation of agitation with no understanding of the underlying cause, compulsory psychiatric care was needed and initiated according to Swedish law (Lag [1991:1128] om psykiatrisk tvångsvård, §3). The brain CT and angiography were normal, and vomiting and a slight increase in temperature were noted. Afterward, the patient described a minor headache attack a week prior to this attack. CSF analysis on day 28 showed a reduced number of leukocytes (Table 1). Treatment with valproic acid 300 mg twice daily was started. A CSF sample at day 28 was analysed for HHV-7 DNA presence using quantitative real-time PCR and was positive, although with low load (1.70, 10-log by the number of genome equivalents/mL, Table 1). Screening for other infectious pathogens in CSF and plasma was negative (Appendix A). HHV-7 PCR was not tested at the onset of HaNDL-like presentation, and viral load at day 28 was too low to conclude if there had been an active infection at the time of diagnosis (Table 1). However, the ICHD-3 diagnostic criteria were not fulfilled for “headache attributed to viral meningitis or encephalitis” (Appendix A) [13]. Headache phenotype differences between the HaNDL and headaches attributed to viral meningitis or encephalitis are presented in Appendix A.

On follow-up three weeks later, no additional headache attacks were registered, and CSF analysis showed a further reduction of leukocytosis. Valproic acid was tapered out, and the patient experienced no further HaNDL-like symptoms. The patient gave written consent to publish anonymized health-related data.

## 3. Discussion

The current case report describes a HaNDL-like presentation with similar clinical signs and pathological findings as previously described, with symptoms including severe headache, aphasia, numbness in extremities [7], increased intracranial pressure, affected consciousness and confusion, CSF pleocytosis, and EEG abnormality changes [2,7].

There are currently no treatment recommendations for HaNDL. However, symptomatic treatment has been applied previously in cases with severe symptomatology: acetazomide to reduce abnormally high intracranial pressure [14], steroidal treatment given against a possible immune-mediated reaction after an infection or other inflammatory triggers [1,15], and treatment with valproic acid thought to affect the possible pathophysiological mechanism of cortical spreading depression (CSD) similar to migraine [16]. Moreover, it is probable that less-known features of valproic acids, such as the capacity to diminish replication of enveloped viruses and anti-inflammatory action through the modulation of innate and adaptive immunity cells [17], could contribute to a beneficial outcome.

Although the brain MRI did not reveal any signs of pathology, the elevated levels of CSF-neurofilament light (NfL; Table 1) suggest an association between brain tissue and axonal integrity damage, in line with cases previously described by Säflund et al. [7], which also identified pathological CSF-NfL levels.

While HHV-7 has a predominant tropism for CD4+ T lymphocytes [18], it exhibits neurotropism and rare but clear involvement in several different neurological diseases, such as Guillain–Barré syndrome, encephalitis, meningitis, demyelinating disorders, and Alzheimer’s disease (AD) [12]. Hence, HHV-7 can be involved in nervous system damage. In our case report, CSF-NfL elevation may have been caused by direct cytopathic effects of virus-infected cells and/or subsequent inflammatory events and autoimmunity following the HHV-7 CNS infection.

Increased levels of chemokine (C-X-C motif) ligand (CXCL) 13 were also seen in CSF, which has not been previously described in a HaNDL patient. The elevated levels of CXCL13 in CSF indicate an implication of B cells in the pathogenesis of HaNDL. CXCL13 is mainly secreted by follicular dendritic cells [19] and is an important chemokine for secondary lymphoid tissue development promoting B cell migration and maturation in germinal centers in response to inflammation or infection. Elevated CXCL13 levels in CSF are associated with CNS infections, such as neuroborreliosis, tick-borne encephalitis virus (TBE), varicella–zoster virus (VZV), and herpes simplex virus (HSV) [20].

The pathophysiology of HaNDL remains unknown. Previous reviews of reported HaNDL cases have suggested that up to 40% exhibit viral-like illness in the weeks prior to developing symptoms [2,3]. Even if this case did not exhibit signs of infection and was not investigated for HHV-7 in the early phase of the illness, it is interesting that this virus, even with a low viral load, was found in CSF. The presence of HHV-7 in this concentration should not be present, so this indicates that the patient recently either had a primary infection with HHV-7 or reactivation the virus; however, we cannot conclude when this happened in relation to HaNDL development. It is possible that the virus was reactivated by increased immunologic activity or that its presence may be involved in the development of this syndrome. All HHVs exhibit at least some degree of neurotropism, and several shares clinical manifestations. As HHV-7 is not included in current routine testing for CNS infections, previous findings of HHV-7 in three HaNDL-like cases, including ours [1,6], warrant including HHV-7 in virus-specific screening panels of CSF with nucleotide acid tests (NAT).

According to the ICDH-3 diagnostic criteria for HaNDL, all aetiological investigations must remain negative [13]. However, this requirement poses a diagnostic challenge in the current case as we cannot determine whether the HHV-7 infection seen was involved in the pathogenesis. Hence, further studies are warranted to determine whether the current and previously described HaNDL cases with CSV HHV-7 findings [1,6] and/or other infectious pathogens described above should be entitled HaNDL or, rather, “HaNDL mimics.”

In search for more specific diagnostic criteria than the current “negative aetiological studies” as described in ICHD-3 and to enable specific and efficient treatment against HaNDL, we urge clinicians to perform broad and sensitive screenings in search for CNS infections to strive for the identification of neurotropic pathogens that may be involved in the pathogenesis of this syndrome. Moreover, diagnostic criteria for HaNDL might require revision as next-generation sequence tools for detecting low concentrations of pathogens might change how we look at actual symptomatic infection or secondary inflammatory activation due to subclinical viral infection.

## 4. Conclusions and Clinical Implications

Improvement in diagnostics: Meta-next-generation sequence tools need to be used to detect pathogens at low concentrations to guarantee HaNDL diagnostic accuracy, according to ICHD-3.

Improvement in treatment: HaNDL treatment by valproic acid and Solu-Medrol could be considered.

## Figures and Tables

**Table 1 pathogens-12-00476-t001:** Dynamic of CSF laboratory findings in time.

Laboratory Findings * (Reference Value)	Day 1	Day 4	Day 28	Day 48
CSF pressure (10–20 cm H_2_O)	35	25	18	N/A
CSF albumin (<260 mg/L)	860	1238	445	356
CSF OCB (pos/neg)	N/A	N/A	neg	N/A
CSF FLC-K (<0.34)	N/A	N/A	0.23	N/A
CSF KFLC (<0.00)	N/A	N/A	−0.31	N/A
CSF leukocytes (<5)–all monocytes	274	385	98	37
CSF CXCL13 (<7.8 ng/L)	N/A	23	62	54
CSF NFL (<380 ng/L)	N/A	N/A	600	570
CSF HHV-7- DNA	N/A	N/A	pos **	N/A

Abbreviations: cm: centimetre; CSF: cerebrospinal fluid; CXCL: chemokine (C-X-C motif) ligand; FLC-K: free light chains type kappa; KFLC: kappa free light chains; L: litre; mg: milligram; N/A: not available; neg: negative; NFL-neurofilament light; OCB: oligoclonal bands. * Comprehensive list of CNS infection-negative screening is presented in Appendix A. ** 1.70, 10-log by number of genome equivalents/mL.

## Data Availability

The datasets generated and/or analyzed during the current study are not publicly available due to patient integrity protection and confidentiality that applies to healthcare information and the rules that apply to access patient data according to Swedish law (25 kap. 1§ Offentlighets- och sekretesslagen [2009:400] and Patientdatalagen [2008:355]).

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
