# Peer review of "Syndrome of Transient Headache and Neurologic Deficits with Cerebrospinal Fluid Lymphocytosis (HaNDL): HHV-7 Finding in Cerebrospinal Fluid Challenges Diagnostic Criteria"

_pathogens, 2023, doi:10.3390/pathogens12030476_

Round 1

Reviewer 1 Report

This case report is devoted to the syndrome of transient headache and neurological deficits with cerebrospinal fluid lymphocytosis (HaNDL), a rare, self-limiting disease with severe headache accompanied by neurological symptoms, for which evidence-based recommendations for diagnosis and treatment are not available due to its rarity and unknown pathophysiology.

The authors describe an interesting case of HaNDL-like disease with HHV-7 DNAemia in cerebrospinal fluid (CSF) in an immunocompetent adult. On arrival at the emergency department, the only abnormal laboratory findings were elevated cerebrospinal fluid pressure, elevated CSF albumin, and a markedly elevated leukocyte count. Other blood and CSF tests which could suggest cerebrospinal inflammation and/or infection were negative. On day 28 of illness, CSF analysis has shown a reduced leukocyte count and the presence of HHV-7 DNA determined by quantitative real-time PCR, but with a low viral load. CFS and plasma screening for other pathogens, including viruses, were negative. Unfortunately, HHV-7 PCR was not tested at the onset (day 1) of HaNDL-like disease, and the viral load at day 28 is too low to conclude whether an active HHV-7 infection was present at diagnosis, which is the main limitation of this interesting and important case report.

Despite this limitation, this case report is essential to encourage clinicians to perform extensive and sensitive screening to identify neurotropic pathogens, including viral infections with low viral load, that may be involved in the pathogenesis of HaNDL.

Minor comments:

Line 18 – “low human herpesvirus 7 (HHV-7) load” instead of “low titers of human herpesvirus 7 (HHV-7)”;

Line 19 – “Low HHV-7 load” instead of “Low titers of HHV-7”;

Line 22 – “with low load” instead of “at low titers”

Line 44 – "A" is redundant - must be removed

Line 92 – “with low load” instead of “at low titer”

Line 95 – “viral load” instead of “levels”

Line 140 – “with low viral load” instead of “at low titer”

Lines-154-155 - It is not clear what the authors mean by "low-grade infections of the CFS", which is not correct in the case of herpesviruses, because they form a lifelong persistent infection and can reactivate.

Table S1 – it is not clear if HSV-1/2 and VZV DNA or antibodies were detected – must be specified.

Reviewer 2 Report

This is a very interesting paper that involves a virus that is not well-known. There are, however, issues when a short paper like this is written. The reader will end up with more quesions than answers.

1)  The herpesvirus is a large and relatively well-known family:

https://www.ncbi.nlm.nih.gov/books/NBK8157/

Many of the viruses are similar in nature in terms of clinical manifestations and cell/irion physiology.

2) As aready mentioned, many of the viruses are similar in nature in terms of clinical manifestations and cell/irion physiology. Many of them are apt at hiding in the body especially the brain and cns. This can be seen HSV (HHV1 and 2) and VZV (Chickenpox). Both viruses stay in the body

https://www.ncbi.nlm.nih.gov/pmc/articles/PMC6861030/

The ability to hide in places like the brain can also be found in the unrelated HIV because HIV and many in the herpes family have highky disordered outer shell that enables the virus to interact more easily with the host proteinsL

https://pubmed.ncbi.nlm.nih.gov/31072073/

The authors may want to mention this along with many of the common chracteristics of the viruses within the herpes family

3) Becasue the hhv family shares similar characteristics including clinical manifestation, a challenge would be how does one distinguish those of HHV-2 with the rest say HSV or VZV. This should be addressed in this paper. Perhaps the authors may want to do a literature search of the symptoms of the vatious HHV and do a comparative analysis

4)There is no mention on how HHV7 is transmitted

Round 2

Reviewer 2 Report

The word "so" in Line 19 is awkwardly used. "such that" is better in this instance